# Establishing a Standardized Clinical Consensus for Reporting Complications Following Lateral Lumbar Interbody Fusion

**DOI:** 10.3390/medicina59061149

**Published:** 2023-06-15

**Authors:** Gregory M. Mundis, Kenyu Ito, Nikita Lakomkin, Bahar Shahidi, Hani Malone, Tina Iannacone, Behrooz Akbarnia, Juan Uribe, Robert Eastlack

**Affiliations:** 1Scripps Clinic Medical Group, San Diego, CA 92037, USA; 2Aichi Spine Hospital, Aichi, Inuyama 484-0066, Japan; 3Mayo Clinic College of Medicine and Science, Rochester, NY 55905, USA; 4San Diego Department of Orthopaedic Surgery, University of California, La Jolla, CA 92093, USAbakbarnia@health.ucsd.edu (B.A.); 5San Diego Spine Foundation, San Diego, CA 92121, USA; 6Barrow Neurological Institute, Phoenix, AZ 85013, USA

**Keywords:** adverse events, minimally invasive spine, circumferential fusion, modified Delphi, OLIF, DLIF, XLIF

## Abstract

*Background and Objectives:* Mitigating post-operative complications is a key metric of success following interbody fusion. LLIF is associated with a unique complication profile when compared to other approaches, and while numerous studies have attempted to report the incidence of post-operative complications, there is currently no consensus regarding their definitions or reporting structure. The aim of this study was to standardize the classification of complications specific to lateral lumbar interbody fusion (LLIF). *Materials and Methods*: A search algorithm was employed to identify all the articles that described complications following LLIF. A modified Delphi technique was then used to perform three rounds of consensus among twenty-six anonymized experts across seven countries. Published complications were classified as major, minor, or non-complications using a 60% agreement threshold for consensus. *Results:* A total of 23 articles were extracted, describing 52 individual complications associated with LLIF. In Round 1, forty-one of the fifty-two events were identified as a complication, while seven were considered to be approach-related occurrences. In Round 2, 36 of the 41 events with complication consensus were classified as major or minor. In Round 3, forty-nine of the fifty-two events were ultimately classified into major or minor complications with consensus, while three events remained without agreement. Vascular injuries, long-term neurologic deficits, and return to the operating room for various etiologies were identified as important consensus complications following LLIF. Non-union did not reach significance and was not classified as a complication. *Conclusions:* These data provide the first, systematic classification scheme of complications following LLIF. These findings may improve the consistency in the future reporting and analysis of surgical outcomes following LLIF.

## 1. Introduction

Lateral lumbar interbody fusion (LLIF) is a minimally invasive operation that employs a lateral retroperitoneal transpsoas approach to the lumbar spine. LLIF is a versatile technique that can be used to treat an array of lumbar pathology and has progressively increased in popularity over the past two decades. This approach offers the ability to create significant lordosis while achieving high rates of fusion [1,2]. However, there are numerous potential risks, including injury to important vascular structures, the bowel, lumbar plexus, and the psoas muscle, amongst others [3,4,5]. Although complication classification schemes exist for the traditional posterior and anterior approaches to the spine [6], none have specifically focused on the unique risks inherent to lateral surgery.

Lateral approaches to the spine necessitate the identification of the psoas and the creation of an operative corridor within the muscle. This allows for a direct pathway to the disc space anterior to the exiting nerve roots, thus facilitating the ability to perform a discectomy and interbody fusion. However, this anatomy has been associated with a unique complication profile when compared to anterior lumbar interbody fusion (ALIF) or transforaminal/posterior lumbar interbody fusion (TLIF/PLIF), respectively. First, the lumbosacral plexus runs directly within the psoas muscle, and injury to these structures remains an important risk during LLIF. In particular, the L4-L5 level has been associated with longer nerve roots [7], and over 60% of patients have been described to report some degree of thigh pain or weakness post-operatively [8,9]. These symptoms have been described to be present for up to 1 year or more after surgery [8]. In addition, lateral approaches to the spine result in the visualization of the vascular anatomy that is distinct from the other techniques. While ALIF can place the great vessels at risk during exposure, LLIF results in the visualization of the segmental arteries that run along the vertebral body [10]. Injury to these vessels can result in substantial bleeding in the setting of a decreased visualization along a narrow operative corridor. Other techniques, such as oblique lumbar interbody fusion (OLIF), have been associated with vascular injuries due to the need to navigate around the iliac vessels during the exposure and mobilization of the vascular sheath away from the psoas [11]. Finally, lateral approaches to the spine necessitate the use of dilators and retraction devices, and posterior retraction along the psoas has been demonstrated to be associated with symptomatic neuropraxia [12].

These unique aspects of lateral surgery thus result in a potential complication profile that is substantially different from the traditional posterior or anterior approaches. Existing large studies have reported an array of complication rates, ranging from 0.59 to 18.0% [13,14], respectively. This variability may be due to the lack of consistency and consensus regarding what constitutes a surgical complication, making the safety and efficacy of this technique difficult to assess when comparing other surgical approaches. Therefore, the purpose of this study was to employ a modified Delphi technique in order to establish a standardized clinical consensus for reporting complications following a LLIF.

## 2. Methods

### 2.1. Review of the Literature

An initial search algorithm was constructed to identify all the manuscripts reporting complications following LLIF within the PubMed/Medline database from 2009 to 2019. This included: “LLIF or lateral lumbar interbody fusion or XLIF or extreme lumbar interbody fusion or DLIF or direct lumbar interbody fusion or OLIF or oblique lumbar interbody fusion” and “complication or adverse event or complications”. The resultant articles were subsequently screened using title and abstract review for relevance to the study question. Manuscripts were fully reviewed and included if they met the following criteria, which were as follows: (a) described a case report or series of patients undergoing a LLIF/DLIF or OLIF, and (b) described the incidence of peri-operative events relative to the study group. Cadaveric studies, as well as manuscripts with combined cohorts and inadequate data to delineate patients undergoing LLIF were all excluded.

### 2.2. Expert Panel

The names of international experts on lateral interbody fusion were obtained from the lists of past international meetings and published studies on lateral access surgery. Twenty-six experts from seven countries were selected by the principal investigators based on the minimum criteria for surgical experience, publication record in the field, and international recognition through conference proceedings. Specifically, experts were considered if they had LLIF surgery experience of more than 100 cases, demonstrated a consistent publication record on the topic of LLIF within the past 10 years, and had presented on LLIF at international meetings.

### 2.3. Survey Consensus/Delphi Technique

Delphi techniques have been widely employed as a research tool in clinical outcomes and social science analyzes in order to resolve problems that cannot be addressed using the available evidence [15,16]. Broadly, these methods revolve around conducting several rounds analyzing specific questions with the selected experts, with the intent of arriving at a consensus. Given the considerable variability in reporting complications following a LLIF, the Delphi technique appeared to be well-suited in providing a standardized framework for the future classification of these events. A cover letter explaining the details of the study was sent to each expert via e-mail. The list of panelists was kept confidential until after the conclusion of the study. Once the participants agreed to be involved in the study, three rounds of surveys were administered through the Google form interface (www.google.com/intl/ja_jp/forms/about/).

Round 1: Participants were asked to categorize each event within the original list of 52 potential complications as a (1) complication, (2) approach-related occurrence, or (3) not a complication, respectively. Based on agreement threshold ranges between 51–80% used in prior Delphi studies [17,18,19,20], a minimum threshold value of 60% was selected to define the consensus in this study [21,22].

Round 2: Based on the results from Round 1, panelists were asked to further classify these complications with the consensus as major or minor. Due to the inherent prevalence and importance of the potential nerve-related outcomes that could have arisen during lateral surgery, events not gaining consensus as complications in Round 1 were hence categorized as either nerve-related or non-nerve related events. For nerve-related events, panelists were asked to further delineate whether (a) the event could be seen as a complication based on the duration of symptoms (<12 weeks or ≥12 weeks), (b) is an expected surgical sequelae, or (c) can be seen as a complication in a unique setting not based on symptom timing (e.g., if it is associated with another complication, necessitates another test/intervention, etc.). Outcomes that were not considered to involve neural injury were then given the opportunity to be re-classified as a (1) major complication, (2) minor complication, or (3) not a complication, respectively.

Round 3. The third round was utilized to evaluate the events that gained consensus as a complication in Round 1 but did not reach consensus as a major or minor complication in Round 2. These outcomes were assessed with additional granularity according to specific peri-operative situations (e.g., cage mal-positioning requiring a return to the OR vs. increased operative time). Nerve-related events that had previously failed to reach consensus were given the opportunity to be reclassified as a major/minor complication based on the symptom duration (Figure 1).

## 3. Results

A total of 52 individual complications were documented [13,14,23,24,25,26,27,28,29,30,31,32,33,34,35,36,37,38,39,40,41,42] from 23 articles that met the inclusion criteria. A panel of 26 experts agreed to participate in the consensus survey representing seven countries, including Argentina (*n* = 1), Australia (*n* = 2), Brazil (*n* = 3), Italy (*n* = 1), Japan (*n* = 2), the United Kingdom (*n* = 1), and the United States (*n* = 16). The entire list of 52 complications was compiled from the literature and reviewed by the panelists.

In Round 1, 41 of the 52 outcomes reached consensus as complications. The greatest consensus (>90%) was achieved for events such as organ injury, fracture, significant infection, vascular injury, and motor nerve deficit. There were six events (thigh pain, thigh numbness, groin pain, groin numbness, intercostal neuralgia, and psoas weakness, respectively) that reached consensus as approach-related complications. However, there were five events (non-union, rupture of the anterior longitudinal ligament (ALL), peritoneal laceration, sensory nerve deficit, and ileus, respectively) that did not reach consensus in any category, as shown in Table 1.

In the second round, 25 of the 41 complications with consensus were classified as major, while 11 were considered as minor, respectively (Table 2). The most significant complications reaching consensus as major events involved motor deficits, spinal fracture, and vascular injury, whereas wound complications, muscle injury, DVT, rib fracture, dural tear, and cage subsidence were agreed upon as minor events. Of the seven nerve-related events that did not reach consensus as a complication in Round 1, thigh pain, groin pain, intercostal neuralgia, and psoas weakness were further classified as a minor complication if they were found to last for ≥12 weeks (Table 3). Of the four non-nerve-related events from Round 1 that did not reach consensus, only ileus reached consensus as a minor complication (Table 4).

In Round 3, events that reached consensus as complications in Round 1, but did not reach major/minor consensus in Round 2 were assessed with increased granularity based on situation-specific addendums where appropriate. Complications requiring the return to the operating room were considered as major, while those that were easily treated with observation, additional clinical evaluation, or pharmacologic management (e.g., antibiotics) were considered as minor (Table 5). Sensory nerve deficits, thigh numbness, and groin numbness were ultimately agreed to be complications if they lasted for a duration of 12 or more weeks (Table 6). Ultimately, 49 of the 52 events were successfully agreed upon for designation as major or minor complications (Table 7).

## 4. Discussion

Post-operative complications remain a crucial component when evaluating surgical approaches to the spine and can serve as an important metric for comparing several different techniques that may be indicated for a specific pathology. Lateral surgery involves a unique operative corridor that increases the propensity for the development of certain complications which are significantly different from the anterior or posterior approaches. These data, which were derived from the evaluation of over 50 published complications by an internationally recognized panel of surgeons provide the first, systematic classification scheme for complications following a LLIF. Our data revealed that vascular/bony/mesenteric injuries, the mal-positioning of instrumentation requiring re-operation, and sensorimotor deficits lasting ≥ 12 weeks achieved high rates of consensus as postoperative complications, while non-union did not reach significance for consensus.

These findings highlight the unique complication profile associated with the lateral approaches to the spine. The vast majority of the existing literature revolves around classifying complications following posterior spinal surgery. For example, Okuda et al. reported data for 251 PLIF patients in an attempt to highlight the intra-operative, early post-operative, and late post-operative adverse events, respectively [43]. The authors found that the majority of complications were associated with dural tears, mal-positioning/failure of instrumentation, and nerve root injury (resulting in radiculopathy, motor/sensory deficits, etc.) In a similar analysis of over 500 patients undergoing MIS-TLIF, Patel et al. described a complication rate of 25.5%, with dural tears, paresthesias, and wound infections being among the most prevalent [44]. In contrast, outcomes of lateral surgery described in the literature predominantly emphasize vascular injuries, bowel-related complications, and sensorimotor deficits [45]. This stems from the substantial differences in the operative corridor, where posterior dissection exposes large portions of the thecal sac, foramen/nerve roots, and subfascial compartments, while lateral approaches involve the direct entry through the psoas and visualization of the iliac/segmental arteries. Notably, 11 of the original 52 complications compiled in our systematic review of the literature involved a sensory or motor event.

Indeed, sensorimotor deficits remain an important and highly prevalent event following lateral surgery. This has been highlighted in numerous studies investigating complications following transpsoas and anterior-to-the-psoas approaches. Cummock et al. examined over 50 patients undergoing transpsoas interbody fusion over a three year period [8]. The authors concluded that 62.7% of patients had thigh symptoms post-operatively, which included dysesthesia, numbness, and weakness. Similarly, Moller et al. reported a prospective series of LLIF patients, and highlighted that up to a quarter of patients experienced anterior thigh numbness and pain [46]. However, 84% experienced complete symptom resolution within six months. Our compiled data corroborates the high prevalence of transient sensorimotor deficits following lateral surgery, with thigh pain/numbness, groin discomfort, psoas weakness, and intercostal neuralgia all identified as potential events. Of note, thigh/groin pain, intercostal neuralgia, and psoas weakness were only agreed upon as complications when lasting 12 weeks or more. As such, future reports of sensorimotor events following LLIF may benefit from additional data on symptom duration, differentiation between the motor/sensory deficits, and the classification of complications for events lasting several months or more.

In the present study, the vast majority (49 of 52) of LLIF-related complications reached classification consensus, including their designation as major or minor events. However, our results for the three events that did not achieve consensus (unplanned ALL rupture, peritoneal laceration, and non-union, respectively) were not unique altogether. These outcomes were considered to be complications when additional surgery was performed, but it was not possible to classify them as major or minor complications. All three came close to being considered as minor (50–55% agreement) but did not meet the 60% criteria for consensus. ALL rupture itself rarely requires treatment, but additional surgery may be necessary if the cage deviates forward. Therefore, while the majority of experts considered ALL rupture to be a minor complication, it could be considered as a major complication in the context of required reoperation. Similarly, while a peritoneal laceration can frequently be closed immediately during surgery, a resultant abdominal hernia could be classified as a major complication if a return to the OR was deemed as necessary. Similar difficulties in reaching consensus regarding these specific complications were encountered in the study by Glassman et al., which focused on complication classification for adult deformity surgery [6]. Between the two studies, motor nerve deficit, cauda equina, vertebral body fracture, pedicle fracture, wound complication, vascular injury, dural tear, and DVT were included in both studies. Of these, vertical body fractures and pedicle fractures were classified as a minor complication by Glassman et al. but were considered to be a major complication in our study. This may represent the divergence of expert opinions related to the approach-related vertebral body injuries that can occur during a LLIF.

Interestingly, non-unions never reached consensus as a post-operative complication. This outcome is particularly challenging to assess, as it has been frequently identified during long-term follow-up, and has been associated with numerous non-surgical patient factors, such as medication use, smoking, and systemic comorbidities [47,48]. As such, it is difficult to ascertain the role that operative factors may play in contributing to pseudarthrosis. Nevertheless, there remains a substantial heterogeneity in the reporting of this outcome, with large numbers of studies describing non-unions separately from the traditional complications and vice versa [49]. Future separation of pseudarthrosis from the other complications may provide a standard benchmark that will facilitate the comparison of both the traditional complications and fusion rates between the various surgical approaches to the lumbar spine.

## 5. Limitations

These findings should be interpreted in the context of limitations in the study design. First, this represents a qualitative synthesis of the existing literature, and is thus inherently limited by weaknesses of the individual studies in question. Several manuscripts did not describe their specific outcomes of interest, or reported compiled data that was difficult to break down into individual complications. We attempted to mitigate this by performing a comprehensive, systematic search in order to incorporate a wide array of articles capturing many different complications. There is certainly a significant need for high quality, prospectively collected data on outcomes following a LLIF. Second, this study relies on the expertise of individual surgeons, who may be influenced by personal experiences and practice patterns. Our goal was to include a diverse group of accomplished experts in order to represent a wide array of views.

## 6. Conclusions

LLIF is associated with a unique complication profile, and these data provide the first, systematic classification scheme for adverse events after surgery. These findings may improve the consistency in the analysis and future reporting of surgical outcomes related to complications following a LLIF.

## Figures and Tables

**Figure 1 medicina-59-01149-f001:**
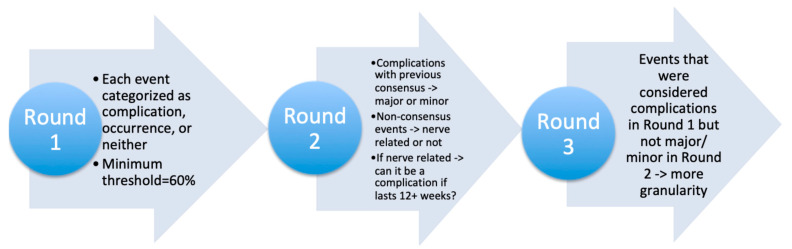
Simplified flowchart depicting the general methodology surrounding the Delphi analysis.

**Table 1 medicina-59-01149-t001:** Round 1: classification of each peri-operative event as a complication, approach-related occurrence, or not a Complication, respectively.

Peri-Operative Event	Complication	Approach- Related Occurrence	Not a Complication
Motor nerve deficit	21 (95.5%)	1 (4.5%)	0 (0%)
Sensory nerve deficit	13 (59.1%)	9 (40.9%)	0 (0%)
Cauda equina deficit	22 (100%)	0 (0%)	0 (0%)
Dural tear	22 (100%)	0 (0%)	0 (0%)
Contralateral nerve palsy	21 (95.5%)	1 (4.5%)	0 (0%)
Thigh pain	2 (8.7%)	21 (91.3%)	0 (0%)
Thigh numbness	2 (8.7%)	21 (91.3%)	0 (0%)
Groin pain	3 (13%)	19 (82.6%)	1 (4.4%)
Groin numbness	4 (18.2%)	16 (72.7%)	2 (9.1%)
Intercostal neuralgia	9 (39.1%)	14 (60.9%)	0 (0%)
Psoas weakness	3 (13%)	20 (87%)	0 (0%)
Male sexual dysfunction	20 (90.9%)	0 (0%)	2 (9.1%)
Retrograde ejaculation	20 (90.9%)	0 (0%)	2 (9.1%)
Vertebral body fracture	22 (95.6%)	1 (4.4%)	0 (0%)
Pedicle fracture	21 (95.5%)	0 (0%)	1 (4.5%)
Cage subsidence	18 (81.8%)	1 (4.6%)	3 (13.6%)
Cage mal-positioning	21 (91.3%)	1 (4.35%)	1 (4.35%)
Cage breakage	20 (86.9%)	1 (4.4%)	2 (8.7%)
Wrong level	17 (70.8%)	0 (0%)	7 (29.2%)
Hardware failure	21 (91.3%)	0 (0%)	2 (8.7%)
Non-union	13 (54.2%)	2 (8.3%)	9 (37.5%)
Unplanned ALL rupture	9 (39.1%)	10 (43.5%)	4 (17.4%)
Iatrogenic disc hernia	21 (95.5%)	0 (0%)	1 (4.5%)
Surgical site infection	20 (95.2%)	1 (4.8%)	0 (0%)
Wound complication	20 (90.9%)	2 (9.1%)	0 (0%)
Major vascular injury	22 (100%)	0 (0%)	0 (0%)
Segmental artery injury	18 (81.8%)	4 (18.2%)	0 (0%)
Superior mesenteric artery syndrome	20 (90.9%)	2 (9.1%)	0 (0%)
Iliac vein laceration	21 (95.5%)	1 (4.5%)	0 (0%)
Lumbar artery pseudoaneurysm	21 (95.5%)	1 (4.5%)	0 (0%)
Epidural hematoma	20 (90.9%)	1 (4.55%)	1 (4.553)
Retroperitoneal hematoma	20 (87%)	1 (4.3%)	2 (8.7%)
Psoas hematoma	14 (60.9%)	7 (30.4%)	2 (8.7%)
Contralateral psoas hematoma	17 (73.9%)	4 (17.4%)	2 (8.7%)
Wound hematoma	15 (65.2%)	6 (26.1%)	2 (8.7%)
DVT	18 (81.8%)	2 (9.1%)	2 (9.1%)
Pulmonary embolism	19 (86.4%)	1 (4.5%)	2 (9.1%)
Rhabdomyolysis	19 (79.2%)	1 (4.2%)	4 (16.6%)
Fascia necrosis	19 (79.2%)	0 (0%)	5 (20.8%)
Rib fracture	15 (62.5%)	8 (33.3%)	1 (4.2%)
Pneumothorax	16 (66.7%)	7 (29.2%)	1 (4.1%)
Lung injury	20 (90.9%)	2 (9.1%)	0 (0%)
Bowel injury	20 (90.9%)	2 (9.1%)	0 (0%)
Ureteral injury	21 (95.5%)	1 (4.5%)	0 (0%)
Sympathetic trunk injury	14 (60.9%)	8 (34.8%)	1 (4.3%)
Diaphragm laceration	15 (62.5%)	9 (37.5%)	0 (0%)
Peritoneum laceration	12 (54.5%)	10 (45.5%)	0 (0%)
Kidney laceration	20 (90.9%)	2 (9.1%)	0 (0%)
Abdominal hernia	20 (90.9%)	2 (9.1%)	0 (0%)
Pseudo hernia	18 (75%)	5 (20.8%)	1 (4.2%)
Ileus	12 (52.2%)	9 (39.1%)	2 (8.7%)
Death	21 (95.5%)	1 (4.5%)	0 (0%)

**Table 2 medicina-59-01149-t002:** Round 2: major/minor classification of events that had previously reached consensus as a complication.

Prior Consensus Complication	Major Complication	Minor Complication
Motor nerve deficit	22 (91.7%)	2 (8.3%)
Cauda equina deficit	24 (100%)	0 (0%)
Dural tear	8 (33.3%)	16 (66.7%)
Contralateral nerve palsy	19 (79.2%)	5 (20.8%)
Male sexual dysfunction	22 (91.7%)	2 (8.3%)
Retrograde ejaculation	20 (82.6%)	4 (17.4%)
Vertebral body fracture	22 (91.7%)	2 (8.3%)
Pedicle fracture	19 (82.6%)	4 (17.4%)
Cage subsidence	3 (12.5%)	21 (87.5%)
Cage mal-positioning	12 (50%)	12 (50%)
Cage breakage	17 (70.8%)	7 (29.2%)
Wrong level	20 (100%)	0 (0%)
Hardware failure	20 (83.3%)	4 (16.7%)
Iatrogenic disc hernia	17 (70.8%)	7 (29.2%)
Surgical site infection	13 (54.2%)	11 (45.8%)
Wound complication	3 (13.6%)	19 (86.4%)
Major vascular injury	24 (100%)	0 (0%)
Segmental artery injury	10 (45.5%)	12 (54.5%)
Superior mesenteric artery syndrome	20 (87.0%)	3 (13.0%)
Iliac vein laceration	22 (100%)	0 (0%)
Lumbar artery pseudoaneurysm	21 (91.3%)	2 (8.7%)
Epidural hematoma	20 (87.0%)	3 (13.0%)
Retroperitoneal hematoma	11 (47.8%)	12 (52.2%)
Psoas hematoma	5 (25%)	15 (75%)
Contralateral psoas hematoma	5 (25%)	15 (75%)
Wound hematoma	2 (10%)	18 (90%)
DVT	9 (37.5%)	15 (62.5%)
Pulmonary embolism	24 (100%)	0 (0%)
Rhabdomyolysis	18 (90%)	2 (10%)
Fascia necrosis	16 (80%)	4 (20%)
Rib fracture	6 (30%)	14 (70%)
Pneumothorax	12 (60%)	8 (40%)
Lung injury	21 (87.5%)	3 (12.5%)
Bowel injury	24 (100%)	0 (0%)
Ureteral injury	24 (100%)	0 (0%)
Sympathetic trunk injury	8 (40%)	12 (60%)
Diaphragm laceration	7 (35%)	13 (65%)
Kidney laceration	24 (100%)	0 (0%)
Abdominal hernia	10 (41.7%)	14 (58.3%)
Pseudo hernia	4 (20%)	16 (80%)
Death	24 (100%)	0 (0%)

**Table 3 medicina-59-01149-t003:** Round 2: further classification of approach-related occurrences according to the duration and peri-operative situation.

Approach-Related Occurrence	Complication when Symptoms < 12 Weeks	Complication when Symptoms ≥ 12 Weeks	Expected Surgical Sequelae	There Is a Situation, Apart from Time, That Would Make You Reclassify from Approach-Related Complication
Sensory nerve deficit	5 (20.8%)	13 (54.2%)	4 (16.7%)	2 (8.3%)
Thigh pain	2 (8.3%)	17 (70.8%)	5 (20.8%)	0 (0%)
Thigh numbness	4 (16.7%)	11 (45.8%)	8 (33.3%)	1 (4.2%)
Groin pain	2 (8.3%)	16 (66.7%)	6 (25%)	0 (0%)
Groin numbness	6 (25.0%)	12 (50%)	6 (25%)	0 (0%)
Intercostal neuralgia	5 (20.8%)	16 (66.7%)	3 (12.5%)	0 (0%)
Psoas weakness	2 (8.3%)	16 (66.7%)	5 (20.8%)	1 (4.2%)

**Table 4 medicina-59-01149-t004:** Round 2: classification of non-neural events from Round 1 that previously did not reach consensus as a complication.

Non-Neural Event	Major Complication	Minor Complication	Not a Complication
Unplanned ALL rupture	4 (20%)	11 (55%)	5 (25%)
Peritoneum laceration	2 (10%)	11 (55%)	7 (35%)
Ileus	1 (5%)	15 (75%)	4 (20%)
Non-union	7 (35%)	10 (50%)	3 (15%)

**Table 5 medicina-59-01149-t005:** Round 3: situation-specific classifications of the events not reaching consensus as a major or minor complication during Round 2.

Event	Major Complication	Minor Complication
Cage mal-positioning: return to the operating room	19 (95%)	1 (5%)
Cage mal-positioning: requiring index surgery repositioning (longer index surgery)	4 (21.1%)	15 (78.9%)
Cage mal-positioning: no repositioning, but additional clinical evaluation	3 (15%)	17 (85%)
Surgical site infection: return to the operating room	18 (90%)	2 (10%)
Surgical site infection: superficial with antibiotic treatment	0 (0%)	20 (100%)
Segmental artery injury: return to the operating room	20 (100%)	0 (0%)
Segmental artery injury: requiring transfusion	14 (70%)	6 (30%)
Segmental artery injury: with intra-operative vascular surgery intervention	18 (90%)	2 (10%)
Segmental artery injury: ligation without sequelae	0 (0%)	20 (100%)
Retroperitoneal hematoma: return to the operating room	20 (100%)	0 (0%)
Retroperitoneal hematoma: requiring IR drainage	16 (80%)	4 (20%)
Retroperitoneal hematoma: treatment with observation	0 (0%)	20 (100%)
Abdominal hernia: return to the operating room	14 (70%)	6 (30%)
Abdominal hernia: consult with general surgery-observation	7 (35%)	13 (65%)

**Table 6 medicina-59-01149-t006:** Round 3: re-classification of nerve-related events that did not reach consensus as complications in Round 2 based on symptom timing.

Nerve-Related Event	Complication Even If Symptoms < 12 Weeks	Complication When Symptoms ≥ 12 Weeks
Sensory nerve deficit	7 (35%)	13 (65%)
Thigh numbness	3 (15%)	17 (85%)
Groin numbness	3 (15%)	17 (85%)

**Table 7 medicina-59-01149-t007:** Overall summary describing the classification of the commonly occurring outcomes of interest.

Outcome of Interest	Final Classification
Dural tear	Minor complication
Sexual dysfunction	Major complication
Pedicle fracture	Major complication
Cage subsidence	Minor complication
Surgical site infection	Major complication if return to OR; minor if antibiotic treatment sufficient
Retroperitoneal hematoma	Major complication if return to OR, or requires IR drainage; minor complication if just observed
Pseudarthrosis	Not a complication (no consensus)
Abdominal hernia	Major complication if return to OR; minor if observed
Psoas weakness	Approach related occurrence if <12 weeks; complication if ≥12 weeks
Thigh numbness	Approach-related occurrence
Ileus	Minor complication
Unplanned ALL rupture	Not a complication (no consensus)

## Data Availability

Data will be made available upon reasonable request to corresponding author.

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
