# Peer review of "Establishing a Standardized Clinical Consensus for Reporting Complications Following Lateral Lumbar Interbody Fusion"

_medicina, 2023, doi:10.3390/medicina59061149_

Round 1

Reviewer 1 Report

This article attempts to define the complications of LLIF using the Delphi method. I think that without higher quality evidence, a Consensus has limited value but can be helpful in facilitating the information flow on certain topics. Therefore, I believe this work can benefit the literature. Methodologically, the article is well written and understandable. I think the tables give a good picture of the various steps in the process, but fail to give an overall view, so I think a summary table of conclusions could be useful. Thank you.

Author Response

Point 1: Therefore, I believe this work can benefit the literature. Methodologically, the article is well written and understandable.

Response 1: Thank you very much for your comments and the time you spent reviewing this submission

Changes 1: No manuscript revisions required for this inquiry.

Point 2: I think a summary table of conclusions could be useful.

Response 2: Thank you very much for your suggestion. We agree. Page 9 now has a summary table depicting the final classification of the most frequently discussed outcomes of interest.

Changes 2: Table 7 now includes an overall summary describing the classification of commonly occurring outcomes of interest.

Reviewer 2 Report

This manuscript presents an effective and systematic study, utilizing a modified Delphi technique to establish a standardized classification for complications following lateral lumbar interbody fusion (LLIF). The international panel of 26 experts from seven countries enhances the credibility of the consensus and ensures diverse perspectives, leading to a more universal applicability of the results.

The authors' strategy of categorizing 52 unique complications into major, minor, or non-complications is an innovative approach that addresses the current lack of consistency in the field. Key complications such as vascular injuries, long-term neurological deficits, and the need for return to the operating room were identified, which may significantly impact clinical decision-making and patient education.

In my view, this manuscript would benefit from one minor revision. It would greatly enhance the paper if the authors could include a table detailing the most prevalent complications along with their incidence percentages as per the papers analyzed. This would provide a concise summary of the key results and allow for a more immediate understanding of the practical implications of the study. With this modification, the manuscript can reach its full potential as a comprehensive and accessible resource in the field of lateral lumbar interbody fusion.

Good job!

Author Response

Point 1: In my view, this manuscript would benefit from one minor revision. It would greatly enhance the paper if the authors could include a table detailing the most prevalent complications along with their incidence percentages as per the papers analyzed. This would provide a concise summary of the key results and allow for a more immediate understanding of the practical implications of the study.

Response 1: Thank you very much for your comments and the time you spent reviewing this submission. We agree. Page 9 now has a summary table depicting the final classification of the most frequently discussed outcomes of interest. This is a more relevant outcome from our data than the percentages in individual papers, as the purpose of this study was to identify a consensus for each outcome.

Changes 1: Table 7 now includes an overall summary describing the classification of commonly occurring outcomes of interest.

Reviewer 3 Report

Mundi et al., presented a comprehensive review of the clinical studies reporting “Complications Following Lateral Lumbar Interbody Fusion” to develop Standardized Clinical Consensus for reporting these complications using modified Delphi technique. Manuscript is written clearly giving details of Methodology and Results. Conclusions are well supported by results.

Author Response

Point 1: Manuscript is written clearly giving details of Methodology and Results. Conclusions are well supported by results.

Response 1: Thank you very much for your comments and the time you spent reviewing this submission.

Changes 1: No manuscript revisions required for this inquiry.

Reviewer 4 Report

Abstract: The conclusion must only include the direct findings of the study. Must be rewritten

Intro: The purpose of the study needs to be clearly mentioned. The current literature evidence on this subject must be elaborated

Methods: Pls present the methodology in  the form of a flow chart

Results: The tables need to be modified for clearer understanding. In the current form, the tables are crowded

Discussion: Pls explain the study purpose in the first para. Highlight the main findings in para 2. The specific topics may be discussed under separate subheadings

Well-written. A few grammatical and typographical errors need to be modified

Author Response

Point 1:  Abstract: The conclusion must only include the direct findings of the study. Must be rewritten

Response 1: Thank you very much for your comments and the time you spent reviewing this submission. The conclusions of the abstract were shortened for clarity.

Changes 1: The conclusion of the abstract now reads: “These data provide the first, systematic classification scheme of complications following LLIF. These findings may improve consistency in the future reporting and analysis of surgical outcomes following LLIF.”

Point 2:  Intro: The purpose of the study needs to be clearly mentioned. The current literature evidence on this subject must be elaborated.

Response 2:  Thank you. We have made sure to clearly describe the purpose of our study at the end of the introduction.

Changes 2:  The final sentence of the introduction on page 2 reads: “Therefore, the purpose of this study was to employ a modified Delphi technique in order to establish a standardized clinical consensus for reporting complications following LLIF.”

Point 3:  Methods: Pls present the methodology in the form of a flow chart

Response 3:  A simplified flowchart depicting the overall Delphi analysis methodology was constructed.

Changes 3:  Page 9 now includes Figure 1, showing a simplified flowchart depicting the general methodology surrounding the Delphi analysis.

Point 4:  Results: The tables need to be modified for clearer understanding. In the current form, the tables are crowded

Response 4:  Thank you very much for your comments and the time you spent reviewing this submission. We agree. Page 9 now has a summary table depicting the final classification of the most frequently discussed outcomes of interest. This is a more relevant outcome from our data than the percentages in individual papers, as the purpose of this study was to identify a consensus for each outcome.

Changes 4:  Table 7 now includes an overall summary describing the classification of commonly occurring outcomes of interest.

Point 5:  Discussion: Pls explain the study purpose in the first para. Highlight the main findings in para 2. The specific topics may be discussed under separate subheadings

Response 5:  Thank you. The purpose of the study, as well as the overall finding, is described.

Changes 5:  Page 10 reads: “These data, which were derived from evaluation of over 50 published complications by an internationally recognized panel of surgeons provide the first, systematic classification scheme for complications following LLIF. Our data revealed that vascular/bony/mesenteric injuries, malpositioning of instrumentation requiring reoperation, and sensorimotor deficits lasting ≥ 12 weeks achieved high rates of consensus as postoperative complications, while non-union did not reach significance for consensus.”

Round 2

Reviewer 4 Report

The recommended changes have been added

The manuscript is acceptable in the current form

The language has good flow throughout the manuscript